# The Number Symbol Coding Task: A brief measure of executive function to detect dementia and cognitive impairment

**James E. Galvin\***, **Magdalena I. Tolea, Claudia Moore, Stephanie Chrisphonte**

Department of Neurology, Comprehensive Center for Brain Health, University of Miami Miller School of Medicine, Miami, Florida, United States of America

\* Jeg200@miami.edu

## Abstract

### Introduction

Alzheimer's disease and related dementias (ADRD) affect over 5.7 million Americans and over 35 million people worldwide. Detection of mild cognitive impairment (MCI) and early ADRD is a challenge to clinicians and researchers. Brief assessment tools frequently emphasize memory impairment, however executive dysfunction may be one of the earliest signs of impairment. To address the need for a brief, easy-to-score, open-access test of executive function for use in clinical practice and research, we created the Number Symbol Coding Task (NSCT).

### Methods

This study analyzed 320 consecutive patient-caregiver dyads who underwent a comprehensive evaluation including the Clinical Dementia Rating (CDR), patient and caregiver versions of the Quick Dementia Rating System (QDRS), caregiver ratings of behavior and function, and neuropsychological testing, with a subset undergoing volumetric magnetic resonance imaging (MRI). Estimates of cognitive reserve were calculated using education, combined indices of education and occupation, and verbal IQ. Psychometric properties of the NSCT including data quality, data distribution, floor and ceiling effects, construct and known-groups validity, discriminability, and clinical profiles were determined.

### Results

The patients had a mean age of 75.3±9.2 years (range 38-98y) with a mean education of 15.7±2.8 years (range 6-26y) of education. The patients had a mean CDR-SB of 4.8±4.7 (range 0–18) and a mean MoCA score of 18.6±7.1 (range 1–30). The mean NSCT score was 30.1±13.8 and followed a normal distribution. All healthy controls and MCI cases were able to complete the NSCT. The NSCT showed moderate-to-strong correlations with clinical and neuropsychological measures with the strongest association (all p's < .001) for measures with executive components (e.g., Judgement and Problem Solving box of the CDR, Decision Making and Problem Solving domain of the QDRS, Trailmaking B, and Cognigram Attention and Executive Composite Scores). Women slightly outperformed men, and

**Data Availability Statement:** A de-identified dataset is available at Open Science Framework DOI (DOI 10.17605/OSF.IO/2CEWK). For questions

regarding this dataset, please contact Michael Kleiman, PhD at mjkleiman@med.miami.edu.

**Funding:** This study was supported by grants from the National Institute on Aging of JEG (R01 AG040211-A1 and R01 NS101483-01A1), the Harry T. Mangurian Foundation, and the Leo and Anne Albert Charitable Trust. The funders had no role in study design, data collection and analysis, decision to publish, or preparation of the manuscript. JEG is the creator of the Number Symbol Coding Task.

**Competing interests:** JEG is the creator of the NSCT. MIT, CM and SC report no conflicts of interest.

individuals with lower educational attainment and lower education-occupation indices had lower NSCT scores. Decreasing NSCT scores corresponded to older age, worse cognitive scores, higher CDR sum of boxes scores, worse caregiver ratings of function and behavior, worse patient and informant QDRS ratings, and smaller hippocampal volumes and hippocampal occupancy scores. The NSCT provided excellent discrimination (AUC: .866; 95% CI: .82-.91) with a cut-off score of 36 providing the best combination of sensitivity (0.880) and specificity (0.759). Combining the NSCT with patient QDRS and caregiver QDRS ratings improved discrimination (AUC: .908; 95% CI: .87-.94).

## Discussion

The NSCT is a brief, 90-second executive task that incorporates attention, planning and set-switching that can be completed by individuals into the moderate-to-severe stages of dementia. The NSCT may be a useful tool for dementia screening, case-ascertainment in epidemiological or community-based ADRD studies, and in busy primary care settings where time is limited. Combining the NSCT with a brief structured interview tool such as the QDRS may provide excellent power to detect cognitive impairment. The NSCT performed well in comparison to standardized scales of a comprehensive cognitive neurology evaluation across a wide array of sociodemographic variables in a brief fashion that could facilitate its use in clinical care and research.

## Introduction

Alzheimer's disease and related dementias (ADRD) currently affect over 5.7 million Americans and over 35 million people worldwide [1]. The number of ADRD cases is expected to increase 3-fold by the year 2050 as the number of older adults is also increasing [1–3]. Community detection of mild cognitive impairment (MCI) and early stages of ADRD is a challenge to clinicians and researchers alike, requiring in-depth evaluations that can be time-consuming. Gold Standard evaluations such as the Clinical Dementia Rating (CDR) [4] and comprehensive neuropsychological testing are used in many research projects but require a trained clinician to administer, interpret, and score the CDR or neuropsychological testing and require an extended period of time with the patient, and in the case of the CDR, an informant. While feasible in research settings (e.g., clinical trials, longitudinal studies), these evaluations may not be practical in primary care settings or epidemiologic case-ascertainment projects [2]. Briefer evaluation tools are often used in these settings. These briefer tools can be grouped into performance-based assessments including the Montreal Cognitive Assessment (MoCA) [5], Mini Mental State Exam [6], or Mini-Cog [7], or interview-based assessments usually with an informant such as the AD8 [8], Informant-Questionnaire in Cognitive Decline in the Elderly [9], or Quick Dementia Rating System (QDRS) [10]. These brief tests are often more heavily weighted towards capturing memory impairment, however other important domains such as attention and executive function may not be well captured. This is unfortunate because alterations in executive problem solving and decision making may be one of the earliest signs of MCI and ADRD [11, 12].

Executive function is a broad construct that captures a number of different aspects including basic functions such as attention, inhibitory control, working memory, set switching, and higher order functions including planning, decision making, and problem solving [13–15].

Several neuropsychological tests characterize executive function and capture declines in individuals with cognitive impairment. Examples include the Digit Symbol Substitution Test from Wechsler Abbreviated Intelligence Scale-Revised (WAIS-R) [16], Stroop Color-Word Test [17] and the Cambridge Neuropsychological Test Automated Battery (CANTAB) [18]. Many of these batteries are lengthy, take expertise to administer, must be interpreted in terms of age and education of the patient, and are proprietary requiring licensing costs. While these tests are in the armamentarium of neuropsychologists, they are not readily accessible to physicians in their office settings or easy to use in community-based research projects. Additionally, some standardized batteries used in large multicenter projects such as the Uniform Data Set (UDS) [19, 20] in the National Institute of Aging Alzheimer Disease Center Program have minimal executive function tasks. To address the need for a brief and easy to score test of executive function for use in clinical practice and in research, we created the Number Symbol Coding Task (NSCT). The goal was to create a brief, valid, easy-to-score, open-access instrument that could discriminate between individuals with and without cognitive impairment capturing attention, problem-solving, and set-switching activities. We examined the NSCT compared with the Gold Standard assessments including the CDR, neuropsychological testing, and neuroimaging.

## Materials and methods

### Study participants

We evaluated 400 consecutive patient-caregiver dyads attending our center for clinical care or participation in cognitive aging research. During the visit, the patient and caregiver underwent a comprehensive evaluation including the Clinical Dementia Rating (CDR) and its sum of boxes (CDR-SB) [4], physical and neurological examination, assessment of mood, physical performance, neuropsychological testing, and caregiver ratings of patient cognitive abilities, behavior, and function. All components are part of standard of care at our center [21]. A waiver of consent was obtained for retrospective review of clinic patients and research participants provided written informed consent. Assent was obtained from all patients. Capacity to consent was determined by a semi-structured interview between the patient and a study clinician. This study was approved by the University of Miami Institution Review Board.

### Development of the Number Symbol Coding Task

The NSCT was developed the lead author and reviewed by the study team. Numbers were selected to represent all of the single digits. Symbols were chosen to be easy to draw through the severe stages of dementia and consisted of simple shapes that could be completed with a maximum of 2 pen strokes. The layout was designed to fit on a single page. The pattern was reviewed by the research team to make sure all numbers and symbols were represented, and that no readily recognizable arrangement could be determined by cognitively healthy controls.

### Administration and scoring of the Number Symbol Coding Task

The NSCT is presented in **Fig 1**. An answer key with 10 numbers corresponding to 10 symbols is provided at the top of the page. Before starting the task, two untimed practice sessions are offered. Practice #1 provides the patient an opportunity to re-code 5 numbers into 5 symbols. If they are able to complete this part, they move on to Practice #2, where they are asked to re-code 5 symbols into 5 numbers. The test then begins with 90 seconds permitted to correctly complete as much of the task as possible. Initially, the patient is presented with a series of numbers to re-code into symbols. In the 17th position, set switching begins with irregular cycles of

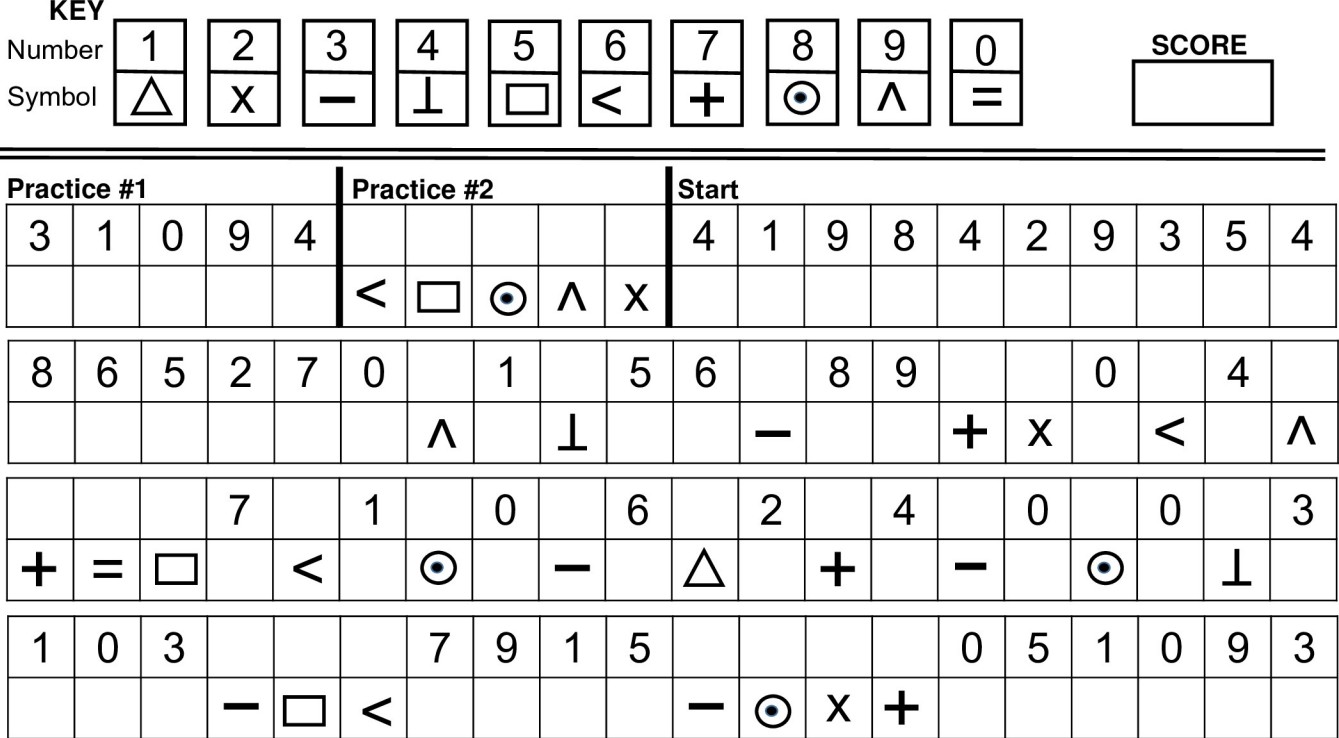

**Fig 1. The Number Symbol Coding Task.**

converting symbols to numbers or numbers to symbols. There are 70 re-coding chances possible, with only correct re-coding counted to give a range of scores 0–70.

## Clinical assessment

The clinical assessments were modelled after the UDS 3.0 [19, 20]. The CDR [4] was used to determine the presence or absence of dementia and to stage its severity; a global CDR 0 indicates no dementia; CDR 0.5 represents MCI or very mild dementia; CDR 1, 2, or 3 correspond to mild, moderate, or severe dementia. The CDR-SB was calculated by adding up the individual CDR categories giving a score from 0–18 with higher scores supporting more severe stages. The Global Deterioration Scale (GDS) [22] was used to provide a global cognitive and function stage: a GDS 1 indicates no impairment; GDS 2 indicates subjective cognitive impairment; GDS 3 corresponds to mild cognitive impairment; GDS 4–7 corresponds to mild, moderate, moderate-severe, or severe dementia [22]. Extrapyramidal features were assessed with the Movement Disorders Society-Unified Parkinson's Disease Rating Scale, motor subscale part III (UPDRS) [23]. The Charlson Comorbidity Index [24] and Functional Comorbidity Index (FCI) [25] were used to measure overall health and medical comorbidities. Global physical performance was captured with the mini Physical Performance Test (mPPT) [26] and frailty was assessed with the Fried Frailty Scale [27]. Vascular contributions to dementia were assessed with the modified Hachinski scale [28]. Consensus diagnoses were determined using standard criteria for MCI [29], AD [30], dementia with Lewy bodies (DLB) [31], vascular contributions to cognitive impairment and dementia (VCID) [32], and frontotemporal degeneration (FTD) [33].

## Estimates of cognitive reserve

Cognitive reserve is a latent moderation construct that represents an individual's ability to maintain cognitive functioning despite the presence of underlying neurodegenerative pathology [34, 35]. While there is no consensus on determinants of cognitive reserve, two of the most important appear to be educational attainment and occupation [35]. Educational attainment was recorded as the number of years of formal schooling. However, the number of years of schooling may not be representative of the quality of the educational experience, and opportunities for advanced education may not be equal across different racial, ethnic and socioeconomic groups [36–40]. The combination of education and occupation was captured by the Hollingshead two-factor index of social status [41], composed of an educational scale (7 levels) and an occupational scale (7 levels) summed to give a social class rating from I-V. This index was used as a proxy for cognitive reserve with Class I representing the highest reserve, II-III representing midlevel reserve, and IV-V representing lowest reserve. Last, verbal IQ was determined with the Test of Premorbid Function (Pearson Assessments, San Antonio, TX) that tests the individual's ability to read a list of 70 words with atypical or irregular grapheme to phoneme pronunciations and presented as tertiles (<100, 100–120, >120).

## Caregiver ratings of patient cognition, function, and behavior

Caregivers completed the informant version of the Quick Dementia Rating System (QDRS) [10] to provide a global rating of cognitive, functional, and behavioral domains. Activities of daily living were captured with the Functional Activities Questionnaire (FAQ) [42]. Dementia-related behaviors were measured with the Neuropsychiatric Inventory (NPI) [43]. Patient daytime sleepiness was assessed with the Epworth Sleepiness Scale (ESS) [44] while daytime alertness was rated on a 1–10 Likert scale anchored by "Fully and normally awake" (scored 10) and "Sleeps all day" (scored 0) [45].

## Cognitive assessment

Each patient was administered a 45-minute test battery modeled after the UDS battery used in the NIA Alzheimer Disease Centers [20] and supplemented with additional measures. The psychometrician was unaware of the diagnosis or CDR. The Montreal Cognitive Assessment [5] was used for a global screen. The rest of the battery included: 15-item Multilingual Naming Test (naming) [20]; Animal naming and Letter fluency (verbal fluency) [20]; Hopkins Verbal Learning Task (episodic memory for word lists–immediate, delayed, and cued recall) [46]; Number forward/backward and Months backwards tests (working memory) [20]; Trailmaking A and B (attention, processing and executive) [47]; the Noise Pareidolia Test [48] (visual perception); and King-Devick Test [49] (visual tracking and saccades). Mood was assessed with the Hospital Anxiety Depression Scale [50] providing subscale scores for depression (HADS-D) and anxiety (HADS-A). Additionally, patients completed the self-reported version of the QDRS [10] for a self-rating of cognitive abilities with scores greater than 1.5 supporting cognitive impairment. The NSCT was administered at the time of the cognitive assessment as an additional measure of executive function.

At a separate sitting, individuals completed the CogState Cognigram Brief Battery (CogState Healthcare LLC, Boston, MA), a well-validated computerized assessment for ages 6–99 years that uses playing cards to test 4 cognitive tasks, providing age normative scores [51, 52]. The Cognigram includes a Detection Task for psychomotor function, an Identification Task for visual attention, a One Card Learning Task for visual memory, and an N-Back Task for working memory. The Cognigram produces 2 composite scores (Memory and Executive).

## Apolipoprotein E genotyping

Apolipoprotein E (ApoE) genotyping was performed by True Health Diagnostics LLC (Richmond, VA). Six possible allelic combinations were obtained with individuals dichotomized as being ApoE 4 carriers or non-carriers.

## Volumetric MRI

A subset of individuals (n = 76) underwent volumetric MRI with NeuroQuant software (CorTechs Labs, San Diego, CA), a FDA-approved automated quantitative analysis of brain MRI images with normative reference data adjusted for age, sex and intracranial volume with high correlation to FreeSurfer [53] and visual assessment [54]. NeuroQuant provides volumes on seven regions of interest: Hippocampus (Bilateral, Right, and Left), Superior and Inferior Lateral Ventricle, Intracranial, Forebrain Parenchyma, Whole Brain, and White Matter Hyperintensities. While hippocampal volume is often used as a predictor of conversion of MCI to AD, hippocampal occupancy (HOC) measures the degree of hippocampal atrophy accounting for volume loss and compensatory inferior lateral ventricle expansion. It is calculated as a ratio of hippocampal volume to the sum of the hippocampal and inferior lateral ventricle volumes in each hemisphere separately, which are then averaged and normalized for age and sex [55]. This measure may aid in differentiation of individuals with congenitally small hippocampi from those with small hippocampi due to a degenerative disorder [55].

**Statistical analyses.** Analyses were conducted with IBM SPSS Statistics v26 (Armonk, NY). Descriptive statistics were used to examine patient and caregiver demographic characteristics, informant rating scales, dementia staging, and neuropsychological testing. One-way analysis of variance (ANOVA) with LSD post-hoc tests were used for continuous data and Chi-square analyses were used for categorical data. To assess scale variability, the frequency distribution, range, and standard deviation were calculated, and data were examined for floor and ceiling effects. Kurtosis and skewness statistics were examined to characterize the shape, symmetry and outliers of the distribution. The NSCT was compared with patient and caregiver characteristics, rating scales, and neuropsychological test performance. Multiple comparisons were addressed using the Bonferroni correction.

Construct validity was assessed comparing the mean performance on each Gold Standard measure with the NSCT using Pearson correlation coefficients [10, 56]. Known-group validity was assessed by examining the NSCT scores by sociodemographic variables and dementia etiology [10, 56]. Receiver operator characteristic (ROC) curves were used to assess the ability of the NSCT to discriminate between individuals with and without cognitive impairment. We first discriminated CDR 0 from CDR >0 and repeated analyses discriminating CDR 0 vs 0.5, which is generally the most difficult staging to determine. The ROC curves were then presented using a potential dementia screening paradigm (a) the NSCT alone, (b) the patient reported QDRS alone, (c) combining NSCT and patient QDRS scores, and finally (d) combining NSCT and patient QDRS scores with the informant version of the QDRS. Results are reported as area under the curve (AUC) with 95% confidence intervals (CIs). Finally, we assessed the ability of the NSCT to differentiate stages of cognitive impairment using mean scores with standard deviations and 95% confidence intervals to provide risk profiles for healthy controls, very mild impairment, mild impairment, and moderate impairment.

# Results

## Sample characteristics

The patients had a mean age of 75.3±9.2 years (range 38-98y) with a mean education of 15.7 ±2.8 years (range 6-26y) of education, and 38.1% were ApoE 4 carriers. Overall, the sample

was 46.9% female with an imbalance of more females (69.2%) in the control group and more males (57.7%) in the cognitively impaired sample ($\chi^2$ = 12.7, p = 0.001). The sample was 97.2% White and 2.3% African American, with 14.5% of the sample reporting Hispanic ethnicity. The patients had a mean CDR-SB of 4.8±4.7 (range 0–18) and a mean MoCA score of 18.6 ±7.1 (range 1–30). The sample covered a range of healthy controls (CDR 0 = 54), MCI or very mild dementia (CDR 0.5 = 161), mild dementia (CDR 1 = 92), moderate dementia (CDR 2 = 64), and severe dementia (CDR 3 = 29). Eighty individuals were unable to perform the NSCT due to cognitive impairment: 0% CDR 0, 0.1% CDR 0.5, 18.5% CDR 1, 57.8% CDR 2, and 82.8% CDR 3. All healthy controls and MCI cases were able to complete the task. Nearly all individuals who were unable to perform the task had ratings of 2 or 3 in the Judgment and Problem Solving domain of the CDR. This gives a final sample size of 320 composed of 53 healthy controls, 120 MCI, 58 AD, 64 DLB, 15 VCID, and 10 FTD cases. The mean NSCT score was 30.1±13.8, with a median of 30. The minimum score was 0 (floor effect: 0.6%) and maximum score was 69 (ceiling effect: 0%) covering nearly the full range of possible scores. Distribution statistics showed skewness was 0.17 (standard error = 0.14) and kurtosis was -0.47 (standard error = 0.27) supporting that the NSCT follows a normal distribution (**Fig 2**).

## Construct validity of the Number Symbol Coding Task with clinical measures

Construct validity is demonstrated in **Table 1** by examining the strength of association between the NSCT and clinical, functional, behavioral, and informant ratings. The NSCT showed moderate-to-strong correlations with clinical measures but most strongly correlated (all p's < .001) with age (R = -.511), FAQ (R = -.583), the informant QDRS (R = -.560), GDS (R = -.715), CDR (R = -.659) and CDR-SB (R = -.724). The NSCT correlated with all CDR domains with the Judgment and Problem Solving box (R = -.743) showing the strongest association and the Personal Care box (R = -.407) showing the weakest association.

## Construct validity of the Number Symbol Coding Task with cognitive performance measures

Construct validity of the NSCT with measures of neuropsychological test performance, mood, and subjective cognitive complaints is shown in **Table 2**. The NSCT showed moderate-to-strong correlations with all neuropsychological tests (p < .001) with the strongest associations with Trailmaking B (R = -.728), MoCA (R = .689), Trailmaking A (R = -.685), Animal Naming (R = .668), and HVLT delayed recall (R = .667). A moderate correlation was were found between the NSCT and the patient QDRS (R = -.473) with Decision Making/Problem Solving (R = -.530) showing the strongest association and Mood (R = -.193) showing the weakest association. The NSCT was not associated with ratings of anxiety or depression. The NSCT was then compared to the Cogstate Brief Battery (Cognigram). There were moderate correlations with the Cognigram Attention (R = .451) and Executive Composite (R = .419) scores.

## Known-groups validity

The performance of the NSCT was compared between patient age, sex, race, ethnicity, education, cognitive reserve, ApoE status, dementia ratings and etiologies in **Table 3**. Females scored higher than males (32.1±14.7 vs 28.1±12.3, p = .03) after controlling for imbalance of sexes in the sample. There was no difference in NSCT scores by race and ethnicity, however given the smaller number of African Americans and Hispanics in the sample, these results should be interpreted with caution. There was a significant difference between all age strata in NSCT scores (all post-hoc p's < .001). There was a significant difference in NSCT by education with post-hoc analyses showing individuals with ≤12 years of education scoring the lowest (post-

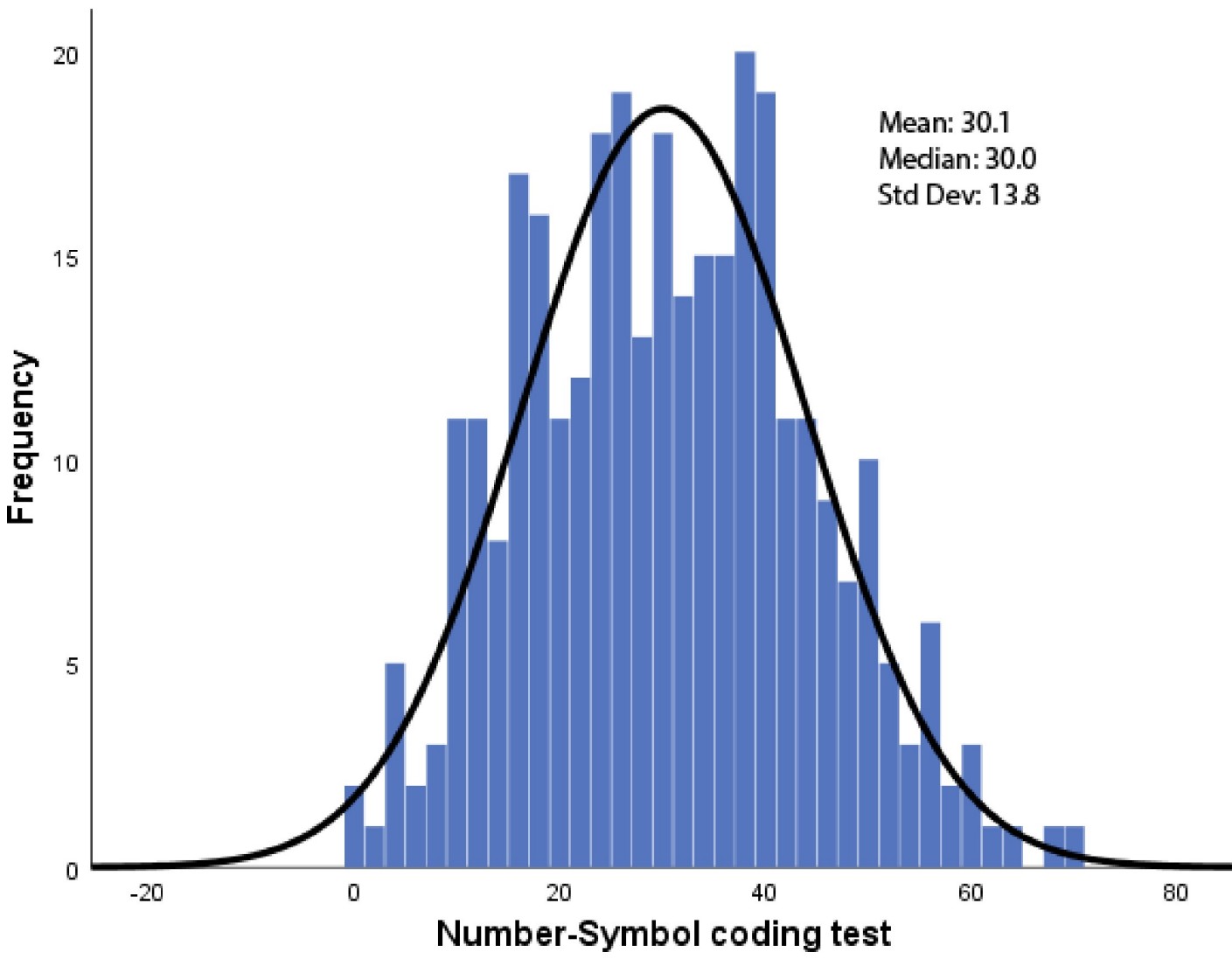

**Fig 2. Histogram of Number Symbol Coding Task.** This histogram demonstrates that the Number Symbol Coding Task follows a normal distribution with a mean of 30.1, standard deviation of 13.8, and a median of 30.

hoc p < .001) and individuals with the lowest cognitive reserve (i.e., education-occupation tertile) scoring the lowest (post-hoc p < .001). However, when examining Verbal IQ strata, there was no difference in NSCT scores. When comparing patient self-ratings of cognitive function with the QDRS, NSCT scores were significantly lower in those with QDRS ≥2 (p < .001). The NSCT total score decreased with higher CDR stages. Post-hoc analyses demonstrate that NSCT scores for each CDR stage is different from all other CDR stages. For GDS stages, GDS 1 (no impairment) and GDS 2 (subjective cognitive impairment) were not different, and NSCT scores decreased across other GDS stages. Post hoc analyses demonstrated that GDS 3 and 4 were different from all other GDS stages, and GDS 5 and 6 were not different from each other. When classifying individuals by consensus clinical diagnoses, NSCT scores in healthy controls were significantly higher than MCI and all dementia etiologies, while MCI individuals were higher than all dementia etiologies (post-hoc p's < .001). Within dementia etiologies, DLB had the lowest NSCT scores.

**Table 1. Construct validity with clinical measures.**

| Variable | R | p-value |
|---|---|---|
| Age | -.511 | < .001 |
| Gender | .147 | .009 |
| Education | .165 | .003 |
| FAQ | -.583 | < .001 |
| NPI | -.380 | < .001 |
| Epworth | -.284 | < .001 |
| Alertness | .358 | < .001 |
| mPPT | .465 | < .001 |
| UPDRS | -.427 | < .001 |
| Charlson | -.240 | < .001 |
| FCI | .228 | .003 |
| Hachinski | -.204 | < .001 |
| Fried | -.460 | < .001 |
| CDR | -.659 | < .001 |
| CDR-SB | -.724 | < .001 |
| GDS | -.715 | < .001 |
| QDRS-Informant | -.560 | < .001 |

Key: FAQ = Functional Activities Questionnaire; NPI = Neuropsychiatric Inventory; mPPT = mini Physical Performance Test; UPDRS = Unified Parkinson's Disease Rating Scale; FCI = Functional Comorbidity Index; CDR = Clinical Dementia Rating; CDR-SB = CDR Sum of Boxes; GDS = Global Deterioration Scale; QDRS = Quick Dementia Rating System.

**Bold** signifies differences after correction for multiple comparisons (corrected p < .003).

## Discriminability of the Number Symbol Coding Task

We tested the ability of the NSCT to discriminate between individuals with and without cognitive impairment using ROC analyses to provide area under the curve (AUC). We first compared healthy controls to individuals with any form of cognitive impairment. The NSCT provided excellent discrimination (AUC: .866; 95% CI: .82-.91) with a cut-off score of 36 providing the best combination of sensitivity (0.880) and specificity (0.759). As detecting the mildest forms of cognitive impairment is the biggest challenge in research and clinical practice, we repeated the analyses to discriminate controls (CDR 0) from those with CDR 0.5 (which includes MCI and very mild dementia). The NSCT provided very good discrimination (AUC: .785; 95% CI: .72-.85). To provide evidence for a brief paradigm for dementia screening, we included the patient reported QDRS as a patient reported outcome, and then combined the patient QDRS with the NSCT (**Fig 3**). The combined battery improved discrimination between healthy controls and impaired individuals (AUC: .890; 95% CI: .85-.93). Lastly, we repeated the analyses adding in the informant version of the QDRS to provide an independent caregiver rating of global cognitive abilities. The addition of the informant QDRS further increased discrimination (AUC: .908; 95% CI: .87-.94). This brief paradigm takes about 5 minutes (patient QDRS: 2–3 minutes, NSCT: 2 minutes) provides excellent ability to detect cognitive impairment. The addition of the informant QDRS when a caregiver is available does not add additional time since the caregiver can independently complete the 2-3-minute QDRS while the patient completes their evaluation.

## Comparison of Number Symbol Coding Task with MRI

We examined the relationship between the NSCT and volumetric MRI performed with Neuro-Quant (**Table 4**). NSCT scores positively correlated with hippocampal volume (R = .528, p <

**Table 2. Construct validity with neuropsychological measures and patient-reported outcomes.**

| Variable | R | p-value |
|---|---|---|
| MoCA | .689 | < **.001** |
| Noise Pareidolia | -.428 | < **.001** |
| Numbers Forward | .242 | < **.001** |
| Numbers Backward | .395 | < **.001** |
| HVLT–immediate | .648 | < **.001** |
| HVLT–delay | .667 | < **.001** |
| HVLT–recognition | .546 | < **.001** |
| Trailmaking A | -.685 | < **.001** |
| Trailmaking B | -.728 | < **.001** |
| Animal Naming | .668 | < **.001** |
| Letter Fluency | .349 | < **.001** |
| MINT | .420 | < **.001** |
| King-Devick | -.579 | .003 |
| AD8 | -.210 | < **.001** |
| QDRS-Patient | -.473 | < **.001** |
| HADS-Anxiety | .016 | .772 |
| HADS-Depression | -.124 | .026 |
| Cognigram-Visual Learning | .353 | .005 |
| Cognigram-Working Memory | .230 | .072 |
| Cognigram-Psychomotor Function | .266 | .024 |
| Cognigram-Attention | .451 | < **.001** |
| Cognigram-Memory Composite | .318 | .012 |
| Cognigram-Executive Composite | .419 | **.001** |

Key: MoCA = Montreal Cognitive Assessment; HVLT = Hopkins Verbal Learning Test; MINT = Multilingual Naming Test; CCI = Cognitive Change Index; CFI = Cognitive Functioning Inventory; QDRS = Quick Dementia Rating System; HADS = Hospital Anxiety and Depression Scale.

**Bold** signifies differences after correction for multiple comparisons (corrected p < .002).

.001), hippocampal occupancy scores (R = .630, p < .001) and inversely correlated with superior lateral ventricle volume (R = -.493, p < .001). There was a stronger relationship between NSCT scores with the left hippocampus than with the right hippocampus. Using the cut-off score of 36 from the ROC analyses, we found significant differences in volume of both hippocampi with a greater difference in the left hippocampus, hippocampal occupancy scores, and in the superior and inferior lateral ventricles. Marginal differences were seen in Forebrain Parenchymal and Whole Brain volumes; however, these are not significant after correction for multiple comparisons.

## Risk profiles

Lastly, to provide a framework for utilizing the NSCT in a clinical setting, we developed a profile of scores by level of impairment based on consensus clinical diagnoses from No Impairment to Moderate Impairment (**Table 5**). Mean NSCT scores and 95% confidence intervals are shown with corresponding global staging by CDR and GDS and patient characteristics. Decreasing NSCT scores correspond to older age, lower MoCA scores, higher FAQ, NPI, CDR-SB, patient and informant QDRS ratings, and smaller hippocampal volumes and hippocampal occupancy scores.

**Table 3. Performance of Number-Symbol Coding Task by demographics, staging, and diagnoses.**

| Sex | | | Race/Ethnicity | | | | |
|---|---|---|---|---|---|---|---|
| **Men** | **Women** | **p-value** | **White** | **Black** | **Hispanic** | **p-value** | |
| 28.1 (12.3) | 32.1 (14.7) | .03[a] | 29.6 (13.7) | 29.9 (12.4) | 32.7 (12.8) | .75 | |
| **Age** | | | | | **ApoE Status** | | |
| **< 60** | **60–69** | **70–79** | **80+** | **p-value** | **Non-Carrier** | **Carrier** | **p-value** |
| 47.8 (12.7) | 35.9 (12.9) | 29.8 (12.3) | 22.5 (10.1) | < .001[b] | 33.9 (13.7) | 30.7 (13.4) | 0.09 |
| **Education** | | | | **SES Class** | | | |
| **≤12** | **13–16** | **>16** | **p-value** | **I** | **II-III** | **IV-V** | **p-value** |
| 24.8 (12.9) | 29.7 (13.5) | 32.3 (13.0) | .002[c] | 32.7 (14.1) | 30.8 (14.8) | 31.9 (13.9) | .69 |
| **Verbal IQ** | | | | **Patient QDRS** | | | |
| **<100** | **100–120** | **>120** | **p-value** | **0–1.5** | **2.0–30.0** | **p-value** | |
| 30.6 (10.4) | 38.6 (14.9) | 32.4 (16.2) | .19 | 36.7 (12.4) | 25.3 (12.5) | < .001 | |
| **CDR** | | | | | | | |
| **0** | **0.5** | **1** | **2** | **3** | **p-value** | | |
| 44.8 (9.8) | 33.4 (10.8) | 19.6 (7.6) | 14.6 (7.9) | 6.0 (6.6)* | < .001[d] | | |
| **GDS** | | | | | | | |
| **1** | **2** | **3** | **4** | **5** | **6** | **p-value** | |
| 46.3 (9.5) | 44.1 (10.3) | 36.2 (9.6) | 22.4 (8.6) | 17.4 (9.7) | 11.9 (7.3)* | < .001[e] | |
| **Diagnoses** | | | | | | | |
| **Control** | **MCI** | **AD** | **DLB** | **VCID** | **FTD** | **p-value** | |
| 45.0 (9.8) | 35.9 (9.4) | 21.6 (11.3) | 17.4 (7.1) | 20.9 (9.7) | 25.3 (12.4) | < .001[f] | |

Mean (SD).

KEY: SES = socioeconomic status; QDRS = Quick Dementia Rating System; CDR = Clinical Dementia Rating; GDS = Global Deterioration Scale; MCI = Mild Cognitive Impairment; AD = Alzheimer's Disease; DLB = Dementia with Lewy Bodies; VCID = Vascular Contributions to Cognitive Impairment and Dementia; FTD = Frontotemporal Degeneration.

Post-hoc Analyses.

[a]Women trend towards better scores than men, controlling for the imbalance in sex between controls and cases.

[b]All age strata are different from each other.

[c]Education ≤12 years different from other education strata.

[d]Each CDR stage different from each other. Note: there are only five CDR 3 individuals able to complete the task.

[e]GDS 1 and 2 are not different from each other; GDS 3 and 4 are different from other stages; GDS 5 and 6 are not different from each other. Note: no GDS 7 individual was able to perform the task.

[f]Controls are different from all other groups; MCI is different from all other groups; Within dementia etiologies DLB is different from AD and FTD, but not VCID.

# Discussion

The NSCT is a brief executive task that incorporates attention, planning and set-switching that can be completed by individuals into the moderate-to-severe stages of dementia. There was very good data quality with a normal distribution and minimal floor and ceiling effects. The NSCT performed equally well across most patient characteristics with women slightly outperforming men and individuals with lower educational attainment and lower education-occupation index scores performing worse. However, there was no difference in NSCT scores with Verbal IQ. There was strong correlation between the NSCT and gold standard measures of cognition, function, and behavior with the strongest association for measures with executive components (e.g., Judgment and Problem Solving box of the CDR, Decision Making and Problem Solving domain of the QDRS, Trailmaking B, and Cognigram Attention and Executive Composite Scores). Number Symbol Coding Task scores declined with greater CDR and GDS staging, and amongst dementia etiologies, individuals with DLB performed worst. A cut-

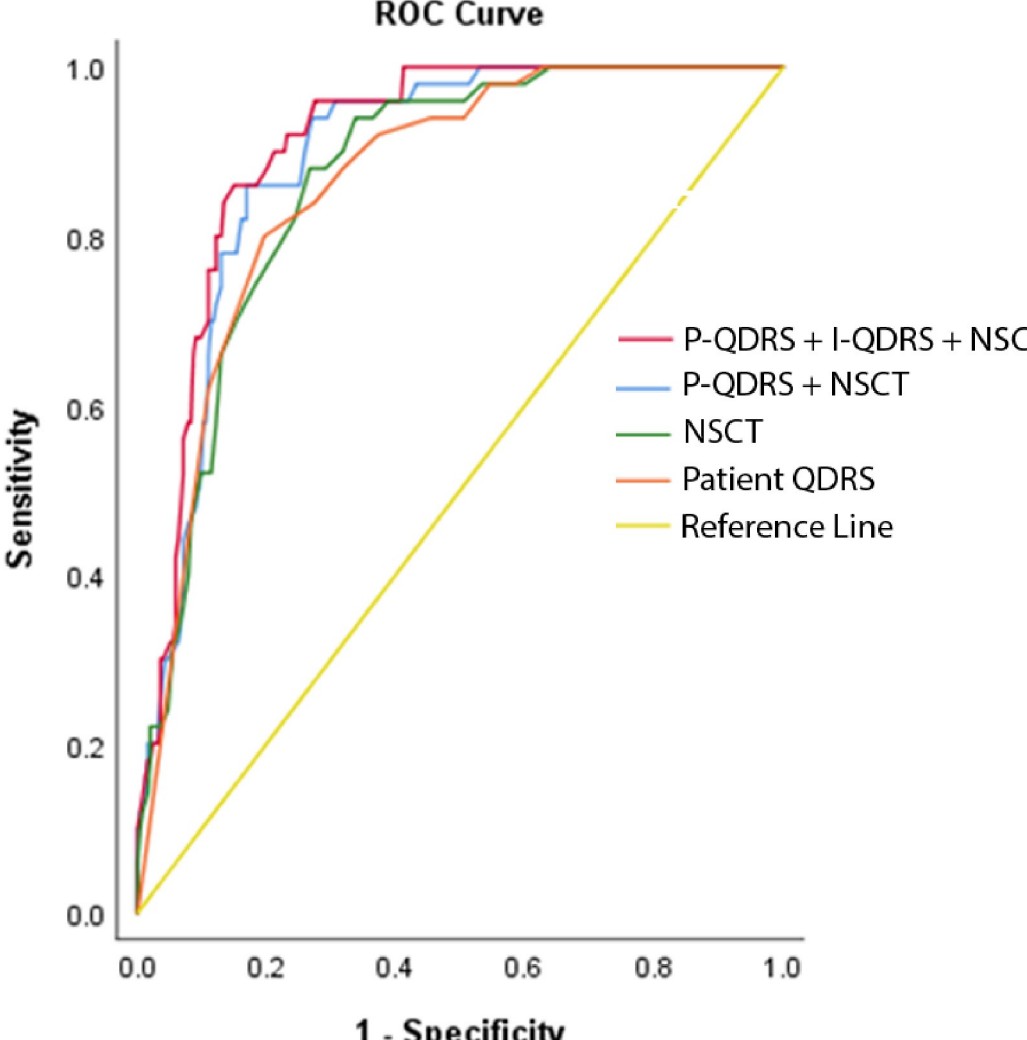

| Comparison | AUC | 95% CI | p-value |
|---|---|---|---|
| QDRS-Patient | .862 | .814-.910 | <.001 |
| NSCT | .866 | .821-.912 | <.001 |
| NSCT + P-QDRS | .891 | .852-.930 | <.001 |
| NSCT + P-QDRS + I-QDRS | .908 | .873-.943 | <.001 |

**Fig 3. Discrimination of the Number Symbol Coding Task.** ROC curves comparing the discriminability of the NSCT, patient and caregiver forms of the QDRS to differentiate healthy controls (CDR 0) from individuals with any form of cognitive impairment (CDR>0). The combination of the NSCT, a patient-reported outcome (patient QDRS), and informant-reported outcome (caregiver QDRS) correctly classified 90.8% of cases (see details in text). Key: AUC = Area under the curve; QDRS = Quick Dementia Rating System; NSCT = Number Symbol Coding Task.

off score of 36 provided the best combination of sensitivity and specificity allowing for the creation of profiles for clinical use. The NSCT correlated with hippocampal and ventricular volumes supporting its relationship with neurodegeneration.

**Table 4. Number Symbol Coding Task and volumetric MRI measures.**

| Brain Region Volume (cm$^3$) | R | p-value | Not Impaired (NSCT>36) | Impaired (NSCT≤36) | p-value |
|---|---|---|---|---|---|
| Hippocampi | .528 | < .001 | 6.9 (1.2) | 5.9 (1.0) | < .001 |
| Left Hippocampus | .515 | < .001 | 3.4 (0.6) | 2.9 (0.5) | < .001 |
| Right Hippocampus | .213 | .067 | 4.0 (2.7) | 2.9 (0.5) | .014 |
| Hippocampal Occupancy Score | .630 | < .001 | 0.77 (0.08) | 0.62 (0.11) | < .001 |
| Superior Lateral Ventricle | -.493 | < .001 | 34.9 (12.6) | 59.9 (31.4) | < .001 |
| Inferior Lateral Ventricle | -.282 | .014 | 2.1 (0.8) | 4.6 (4.7) | **.006** |
| Intracranial | .059 | .618 | 1516.3 (154.8) | 1476.6 (271.3) | .477 |
| Forebrain Parenchyma | .272 | .018 | 957.4 (113.6) | 875.4 (164.3) | .021 |
| Whole Brain | .265 | .022 | 1109.8 (127.2) | 1020.2 (185.9) | .026 |
| White Matter Hyperintensities | .088 | .705 | 9.1 (22.3) | 5.8 (8.3) | .647 |

Mean (SD).

**Bold** signifies differences after correction for multiple comparisons (corrected p < .00625).

Executive dysfunction in MCI and early stage ADRD may be an important but overlooked construct. Executive dysfunction may have significant impact on activities of daily living and quality of life, perhaps more so than memory impairment [57–60]. We previously demonstrated transitions in performance on traditional executive and attention tasks such as Trail-making A and B, Block Design, and Visual Retention tests [61] occur up to three years prior to clinical diagnosis of MCI and dementia in individuals who eventually developed AD dementia [12] and Parkinson's disease dementia [62, 63]. Furthermore, in healthy controls who came to autopsy with evidence of preclinical AD, judgement and problem solving (CDR box domain)

**Table 5. Clinical profiles of Number Symbol Coding Task scores.**

| | No Impairment | Very Mild Impairment | Mild Impairment | Moderate Impairment | |
|---|---|---|---|---|---|
| NSCT Mean (SD) | 44.8 (9.8) | 33.4 (10.8) | 19.6 (7.6) | 14.6 (7.9) | |
| NSCT 95% CI | 42.1–47.5 | 31.7–35.0 | 17.9–21.4 | 11.5–17.8 | |
| Corresponds to CDR | 0 | 0.5 | 1 | 2 | |
| Corresponds to GDS | 1/2 | 3 | 4 | 5 | |
| **Patient Characteristics** | | | | | p-value |
| Age | 67.7 (10.9) | 74.2 (8.6) | 77.8 (7.9) | 78.1 (7.8) | < .001 |
| MoCA | 26.6 (2.4) | 22.1 (3.6) | 17.3 (4.6) | 13.5 (4.9) | < .001 |
| CDR-SB | 0.1 (0.2) | 1.8 (1.2) | 5.4 (1.4) | 9.6 (1.5) | < .001 |
| FAQ | 0.1 (0.5) | 3.6 (4.8) | 12.3 (6.9) | 20.8 (7.2) | < .001 |
| NPI | 1.4 (1.9) | 5.1 (4.3) | 8.7 (6.1) | 9.9 (5.4) | < .001 |
| QDRS-Patient | 0.5 (1.0) | 2.8 (2.7) | 5.8 (4.8) | 8.5 (4.7) | < .001 |
| QDRS-Informant | 0.7 (1.0) | 3.4 (3.1) | 7.8 (3.9) | 12.4 (4.2) | < .001 |
| Hippocampal Volume (cm$^3$) | 7.4 (1.2) | 6.2 (1.1) | 5.7 (1.0) | n/a[1] | **.002** |
| Hippocampal Occupancy Score | 0.78 (0.11) | 0.68 (0.11) | 0.58 (0.09) | n/a[1] | **.001** |

Mean (SD).

Key: NSCT = Number Symbol Coding Task; CDR = Clinical Dementia Rating; CDR-SB = CDR Sum of Boxes; GDS = Global Deterioration Scale; MoCA = Montreal Cognitive Assessment; FAQ = Functional Activities Questionnaire; NPI = Neuropsychiatric Inventory; QDRS = Quick Dementia Rating System.

[1]Volumetric MRIs not conducted in moderate dementia individuals.

**Bold** signifies differences after correction for multiple comparisons (corrected p < .006).

and attention tasks (Trailmaking A) were among the first domains to exhibit clinically detectable change [11]. Similar findings of early executive dysfunction have been reported in other studies of ADRD [18, 58, 64, 65]. The challenge for clinicians is that most brief testing instruments have limited ability to test executive function, and when present, are limited to shortened versions of traditional attention-executive tasks such as a brief version of Trailmaking B in the MoCA [5] or a Clock Drawing in the Mini-Cog [7].

Complexity in testing is more likely to bring out deficits with an inability to compensate in individuals with neurodegenerative diseases, particularly if the task taps into basic (e.g., attention, inhibitory control, set switching) and higher order (e.g., planning, problem solving) functions [13]. The NSCT offers this complexity especially with switching back and forth from number-to-symbol coding to symbol-to-number coding. Individuals with MCI and ADRD have significant slowing down of the coding task compared with healthy controls. This set-switching component further differentiates the NSCT from other executive tests such as Digit Symbol Substitution of the WAIS-R [16].

There are several limitations in this study. The NSCT was validated in the context of an academic research setting where the prevalence of MCI and dementia are high, and the patients tend to be highly educated and predominantly White. Validation of the NSCT in other settings where dementia prevalence is lower (i.e. community samples) and the sample is more diverse is needed. As this is a cross-sectional study, the longitudinal properties of the NSCT still need to be elucidated. The clinical profile and cut-off scores were developed in this convenience sample and presented as a guide. Future studies should include a more diverse community population. The majority of cases consisted of MCI, AD, and DLB with fewer VCID and FTD cases. Biomarker examination was limited to ApoE genotypes and MRI. Although NeuroQuant and more commonly used research programs for volumetric analyses (i.e., Freesurfer) are similar [53], the number of regions available from NeuroQuant are limited. This is especially true for analysis of individual cortical volumes as executive tasks are traditionally associated with frontal lobe functioning. Analyses with other specific biomarkers such as amyloid β-protein, tau, or α-synuclein are needed.

Strengths of this study include the use of a comprehensive evaluation that is part of standard of care with extensive characterization of patients and measurement of cognitive, functional, and behavioral constructs using Gold Standard instruments. Another advantage of the NSCT is its brevity being completed in 90 seconds and easy scoring of counting only correct re-coding. The NSCT could be administered regardless of dementia etiology and can be completed by patients through the moderate-to-severe stages of dementia. The NSCT may be a useful tool for dementia screening, case-ascertainment in epidemiological or community-based ADRD studies, and in busy primary care settings where time is limited. Combining the NSCT with a brief structured interview tool such as the QDRS may provide excellent power to detect cognitive impairment. Patients or research participants could then be referred for a more extensive evaluation. The NSCT performed well in comparison to standardized scales of a comprehensive cognitive neurology evaluation across a wide array of sociodemographic variables in a brief fashion that could facilitate its use in clinical care and research.

## Author Contributions

**Conceptualization:** James E. Galvin.

**Data curation:** James E. Galvin, Magdalena I. Tolea, Claudia Moore, Stephanie Chrisphonte.

**Formal analysis:** James E. Galvin.

**Funding acquisition:** James E. Galvin.

**Methodology:** James E. Galvin.

**Project administration:** Stephanie Chrisphonte.

**Supervision:** James E. Galvin.

**Writing – original draft:** James E. Galvin.

**Writing – review & editing:** James E. Galvin, Magdalena I. Tolea, Claudia Moore, Stephanie Chrisphonte.

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
