## [Decision Letter · Decision Letter 0]

29 Oct 2020

The Number Symbol Coding Task: A Brief Measure of Executive Function to Detect Dementia and Cognitive Impairment

PONE-D-20-27461

Dear Dr. Galvin,

We’re pleased to inform you that your manuscript has been judged scientifically suitable for publication and will be formally accepted for publication once it meets all outstanding technical requirements.

Kind regards,

Linda Chao

Academic Editor

PLOS ONE

1. Please describe in your methods section how capacity to consent was determined for the participants in this study.

2.Please provide additional details regarding participant consent. In the ethics statement in the Methods and online submission information, please ensure that you have specified (1) whether consent was informed and (2) what type you obtained (for instance, written or verbal, and if verbal, how it was documented and witnessed). If your study included minors, state whether you obtained consent from parents or guardians. If the need for consent was waived by the ethics committee, please include this information.

Reviewers' comments:

Reviewer's Responses to Questions

**Comments to the Author**

1. Is the manuscript technically sound, and do the data support the conclusions?

Reviewer #1: Yes

Reviewer #2: Yes

2. Has the statistical analysis been performed appropriately and rigorously? 

Reviewer #1: Yes

Reviewer #2: Yes

3. Have the authors made all data underlying the findings in their manuscript fully available?

Reviewer #1: Yes

Reviewer #2: Yes

4. Is the manuscript presented in an intelligible fashion and written in standard English?

Reviewer #1: Yes

Reviewer #2: Yes

5. Review Comments to the Author

Reviewer #1: The Number Symbol Coding Task: A Brief Measure of Executive Function to Detect Dementia and Cognitive Impairment – Review for PLOS ONE

The authors have developed the Number Symbol Coding Test (NSCT), a brief, easy-to-score, measure of executive functioning with the goal to detect the earliest signs of executive dysfunction. As they note, changes in executive function may emerge even earlier than memory changes and can be a strong indicator of early functional change in IADLs and ADLs, justifying the utility of a measure that can be quickly administered in primary care offices or for research protocols. This assessment is an important contribution to both the literature and clinical practice, and it is quite timely given the growing population of older adults with MCI and dementia.

Additional strengths include the good construct validity for functional ability in discriminating between CDR of 0 and 0.5. Discriminability improved with the addition of a brief questionnaire (Quick Dementia Rating System; QDRS), which the authors suggest combining with the NSCT for a brief, 5-minute paradigm that provides increased precision in detecting early cognitive impairment. Given these strengths, there are several areas the authors may consider to further strengthen their paper and reflect on for future research:

1) The authors found that the NSCT correlated most with Trails B, MoCA, and Trials A, demonstrating good convergent validity to tests of executive functioning and psychomotor processing speed. In future research, they may want to consider demonstrating divergent validity by comparing the NSCT to the SDMT and WAIS-IV Coding subtest to confirm that the set-shifting component of the NSCT does in fact tap into executive functioning beyond the level of these similar tests.

2) A cutoff score of 36 on the NSCT was found to have the best discriminability and is suggested for use across all ages, levels of education, and racial groups. The need for good threshold cutoff scores, particularly for a brief screening measure, can certainly be appreciated in clinical settings. However, this can be limiting for minority groups or individuals with lower education. Given the nature of the convenience sample in the current study, creating norms would be problematic. The authors may consider gathering additional data in future studies with a larger, more racially diverse sample to create norms and/or specified cutoff scores with good sensitivity and specificity for various demographic groups.

3) It would be interesting for the authors to provide (briefly) additional information about how the test was developed (i.e., how decisions were made about the pattern, new symbols, layout, etc.) and if any other versions of the test were trialed.

4) Participants with LBD performed worst on the NSCT in comparison to all other groups. Given the deficits seen in LBD, this raises the concern regarding how motor dexterity was controlled for.

5) Traditionally, neuropsychologists would expect an executive functioning task to be primarily corrected with the frontal lobes; however, this study found that poorer performance on the NSCT was most highly correlated with left hippocampal volume loss. It would be helpful for the authors to comment on whether they believe this finding is indicative of the NSCT detecting preclinical Alzheimer’s disease or merely a suggestive of a non-pathological verbal memory deficit.

Reviewer #2: This was a pertinent methodology paper of wide interest for researchers and clinicians in the field of dementia. It is imperative to make such cognitive instruments available for wider testing and validation in different settings.

Please revise Figure 3: Discrimination of the Number Symbol Coding Task. There is ambiguity between diagram vs table: curves represented in the diagram show best AUC in NSCT followed by P-QDRS- this is in contrast to figures in the table under the diagram showing best AUC @.908 for combo NSCT + P-QDRS + I-QDRS!

6. PLOS authors have the option to publish the peer review history of their article (what does this mean?). If published, this will include your full peer review and any attached files.

Reviewer #1: No

Reviewer #2: **Yes: **Simona Sacuiu

---

## [Editor Report · Acceptance letter]

17 Nov 2020

PONE-D-20-27461 

The Number Symbol Coding Task: A Brief Measure of Executive Function to Detect Dementia and Cognitive Impairment 

Dear Dr. Galvin:

I'm pleased to inform you that your manuscript has been deemed suitable for publication in PLOS ONE. Congratulations! Your manuscript is now with our production department. 

Kind regards, 

on behalf of

Dr. Linda Chao 

Academic Editor

PLOS ONE